environmental chemistry/environmental science/analytical chemistry

cigarette smoke, aldehydes and ketones in particle phase, Girard T reagent, FT-ICR MS, Orbitrap MS

**Authors for correspondence:**
Xibin Zhou
e-mail: dawei492@aliyun.com
Xuezheng Liu
e-mail: lxz1962jmu@aliyun.com

This article has been edited by the Royal Society of Chemistry, including the commissioning, peer review process and editorial aspects up to the point of acceptance.

# Molecular characterization of aldehydes and ketones in particle phase of mainstream and sidestream cigarette smoke

Xiu Chen[3], Quan Shi[3], Xibin Zhou[1] and Xuezheng Liu[2]

[1]College of Basic Science, and [2]School of Basic Medical Sciences, Jinzhou Medical University, Jinzhou, Liaoning 121001, People's Republic of China
[3]State Key Laboratory of Heavy Oil Processing, China University of Petroleum, Beijing 102249, People's Republic of China

XZ, 0000-0002-8252-2405

Aldehydes and ketones (AKs) in cigarette smoke are risk to humans and environment. Due to the complexity of itself and the interference of the smoke tar matrix, the aldehydes and ketones in particle phase (AKPs) of mainstream smoke (MSS) and sidestream smoke (SSS) have not been well investigated. In this study, the AKPs of MSS and SSS were derivatized into polar products by reaction with Girard T reagent. The derivatives were isolated rapidly by column chromatography and analysed by Fourier transform ion cyclotron resonance mass spectrometry (FT-ICR MS). Fifteen species of aldehydes and ketones were detected by positive ion electrospray ionization (ESI) FT-ICR MS: $O_{1-6}$, $N_1O_{1-4}$, $N_2O_{1-3}$ and $N_3O_{2-3}$. The total number of AKPs obtained by ESI FT-ICR MS in MSS and SSS is about 1100 and 970, respectively. After hydrolysis, the original AKPs were obtained and 63 carbonyls were identified and quantified by gas chromatography–mass spectrometry (GCMS). The nitrogen-containing and high-oxygen AKPs were further characterized by Orbitrap mass spectrometry. Structures of compounds with high relative abundance in the mass spectrum were speculated (e.g. a series of degradants of cembrenediol) by comparison with the results of GCMS.

## 1. Introduction

Tobacco constituents and additives produce a large number of aldehydes and ketones (AKs) during combustion [1–4]. The AKs have the potential to induce endogenous oxidative stress

and inflammation [5]. Some AKs in tobacco smoke have been classified as probably or possibly carcinogenic to humans by the International Agency for Research on Cancer [6]. AKs from tobacco smoke may irritate the respiratory system of non-smokers and induce asthma in children and have even been linked with leukaemia [7–9]. AKs in tobacco smoke contribute to the development of smoking-related diseases [10]. Therefore, analysis of AKs in cigarette smoke is critical for assessing human and environmental risks.

The AKs in the complex matrix can be directly analysed by gas chromatography–mass spectrometry (GCMS) [11,12] or other technique. Recently, Zimmermann and other groups [13–22] have developed a series of rapid and sensitive non-targeted mass spectrometry techniques to comprehensively analyse the carbonyls and other components in complex matrix. The non-targeted method has the advantage that there are no complicated post-processing steps, and it is fast and accurate. Other than these, chemical derivatization is a very attractive method due to its selectivity [23–25]. Derivatization of AKs has two purposes: (i) AKs readily react with nucleophile or oxidant, so a protection of the AKs prior to analysis is advantageous, and (ii) derivatization is helpful for identification, isolation and purification in complex matrices. Therefore, a large number of derivatization reagents for AKs have been used.

Commonly used derivatization reagents include hydroxylamine [26], 2,4-dinitrophenylhydrazine (DNPH) [27], Girard's reagent T [28–30], 2-hydrazino-1-methylpyridine (HMP) [31], cyclohexanedione [32], 4-[2-(trimethylammonio)ethoxy] benzen-aminium halide (4-APC) [33] and N,N-dicarboxymethyl hydrazine (DCMH) [34]. The Girard reagent is an old but very useful reagent for chemoselective derivatization of carbonyl compounds [30,35,36]. It has been widely used for the separation and identification of AKs in nature products [37], proteins [38], fossil fuel [28], etc. By derivatization, several AKs have been identified by GCMS in cigarette smoke. However, due to the complexity of itself and the interference of the smoke tar matrix, the application of Girard reagent to the analysis of AKs in particle phase of cigarette smoke is rarely performed. In addition, the GC and liquid chromatography (LC) are limited by the operation temperature or the separation ability, the AKs with high boiling point and complex structure in particle phase cannot be effectively separated and characterized [39]. Electrospray ionization (ESI) is a common ionization technique used in modern MS, which ionizes polar compounds and is not limited by boiling point [39–41]. The high resolution mass spectrometry (such as Fourier transform ion cyclotron resonance mass spectrometry (FT-ICR MS) and Orbitrap MS) were used for the molecular characterization of super-complex fossil fuels [28,40,42–45]. The ESI coupled with high resolution mass spectrometry is expected to provide more details on the molecular composition of aldehydes and ketones in the particle phase (AKPs) of cigarette smoke.

Girard reagent can react with AKs to form cationic derivatives. The pre-charged property of Girard reagent can enhance the ionization efficiency of the derivatives in ESI-MS analysis [46]. Recently, a method of Girard reagent derivatization followed by high resolution MS has been successfully used for the analysis of AKs in low-temperature coal tar. Moreover, these derivatives can be easily separated by column chromatography and then converted to the original AKs by hydrolysis in high yield [29].

Cigarette smoke include sidestream smoke (SSS, the smoke emanating between puffs) and mainstream cigarette smoke (MSS, the smoke inhaled by the smoker) [47]. In this work, the AKPs of MSS and SSS were derivatized by reaction with Girard's reagent T, respectively. The derivatives were analysed by ESI FT-ICR MS. After hydrolysis, the original carbonyl compounds were obtained and then characterized by GCMS and ESI Orbitrap MS in detail.

# 2. Material and methods

## 2.1. Reagents and cigarettes

Girard's reagent T (GirT), anhydrous sodium sulphate ($Na_2SO_4$) and potassium hydroxide (KOH) were purchased from Aladdin Ltd (Shanghai, China). Analytical grade methanol (MeOH), dichloromethane (DCM), tetrahydrofuran (THF) were purchased from Tianjin yongda Chemical Reagents Company (Tianjing, China) and distilled before use. Silica gel (100–200 mesh) was purchased from Qingdao Haiyang Chemical Company (Qingdao, China) and purified by Soxhlet extraction with $CH_2Cl_2$ for 24 h before use. Acetic acid and hydrochloric acid were obtained from Tianjin yongda Chemical Reagents Company and used as received without further purification. Commercially available brand (Huangshan Xinyipin) of cigarettes was used for this study.

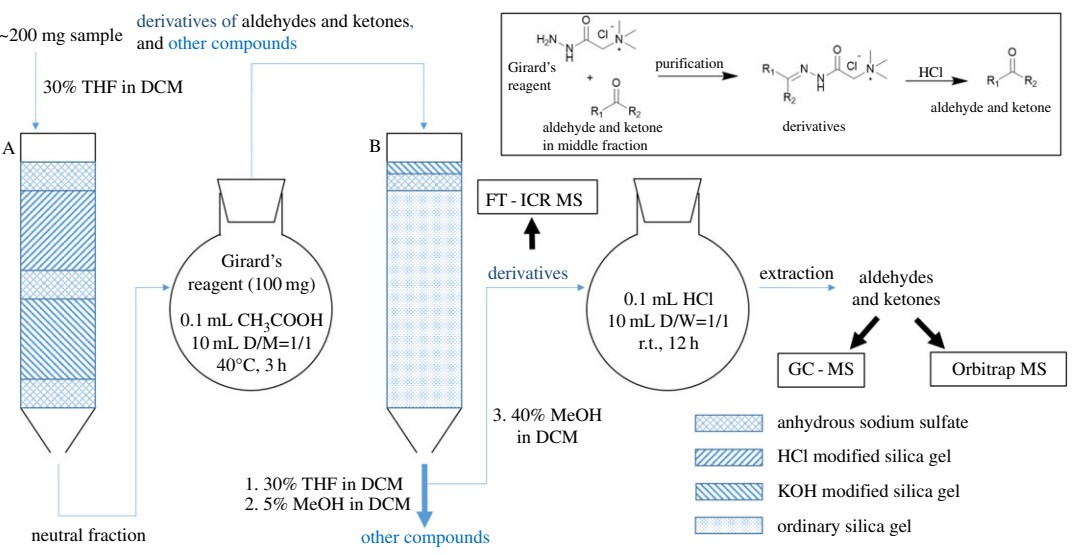

**Figure 1.** Separation-reaction-analytical test scheme for the AKPs.

## 2.2. Sample preparation

Before smoking, the cigarettes were conditioned at 22°C and 60% humidity for 24 h. Cigarettes were smoked according to the ISO 4387 smoking procedure: 1 puff min$^{-1}$ with a 2 s puff duration, and a 35 ml puff volume. The fishtail chimney device were used to collect the particle phase of SSS. Twenty cigarettes were divided into four groups on average. The particulate phase of MSS and SSS of each group was collected on fibre pads under a set of internationally agreed standard smoking conditions. The particle phase on four fibre pads was eluted by DCM/MeOH = 3/1, combined and stored at −20°C.

## 2.3. Derivatization and separation of aldehydes and ketones

The original AKs are neutral compounds, so in the first step the high-polarity acidic and basic fractions were removed from the sample. The HCl modified silica gel [48] and KOH modified silica gel [49,50] have been used to obtain basic and acidic compounds from fossil fuels. In this work, the HCl modified gel and KOH modified gel were combined in one chromatography column, to remove high-polarity acidic-basic fractions and obtain the neutral fraction in one step. This solid phase extraction (SPE) method can avoid emulsification during liquid–liquid extraction and greatly save operating time. In this step, the dichloromethane (DCM)/tetrahydrofuran (THF) = 7/3 (v/v) mixture was used as the eluent to elute the neutral fraction. The polarity of DCM/THF = 7/3 (v/v) is high enough to elute the aldehydes and ketones. In the second step, the neutral fraction mixed with Girard reagent and the AKs are converted into corresponding GirT derivatives. The resulting GirT derivatives are very easy to separate. This is because the polarity of the quaternary ammonium salt is much higher than the other components in the neutral fraction. The mixture was subjected to the chromatography and the DCM/methanol = 20/1 (v/v) was used as the eluent. Because the polarity of DCM/methanol = 20/1 (v/v) is much higher than the DCM/THF = 7/3 (v/v), the other neutral compounds were removed very cleanly. The GirT derivatives still retain on the silica gel and cannot be eluted due to its high polarity. Then a much higher polarity eluent, DCM/methanol = 3/2 was used to elute the GirT derivatives. The huge difference in polarity between these two parts ensures only AKs are present in the AKs fraction. The detailed process is as follows and shown in figure 1.

One gram of anhydrous sodium sulphate, 2 g of KOH-modified silica gel, 1 g of anhydrous sodium sulphate, 4 g of HCl acid-modified silica gel and 1 g anhydrous sodium sulphate were sequentially packed into a 1 cm diameter column A. Ten grams of ordinary silica gel, 1 g of anhydrous sodium sulphate and 0.5 g of KOH-modified silica gel were sequentially packed into another 1 cm diameter column B. The obtained cigarette smoke particle phase (shown in table 1) was dissolved in 4 ml of a DCM/THF = 7/3 (v/v) mixture, and the solution was subjected to the chromatography A using 25 ml of a DCM/THF = 7/3 (v/v) mixture as the eluent to elute the neutral fraction. The polarity of the mixed solvent is high enough to elute the aldehydes and ketones in the neutral fraction.

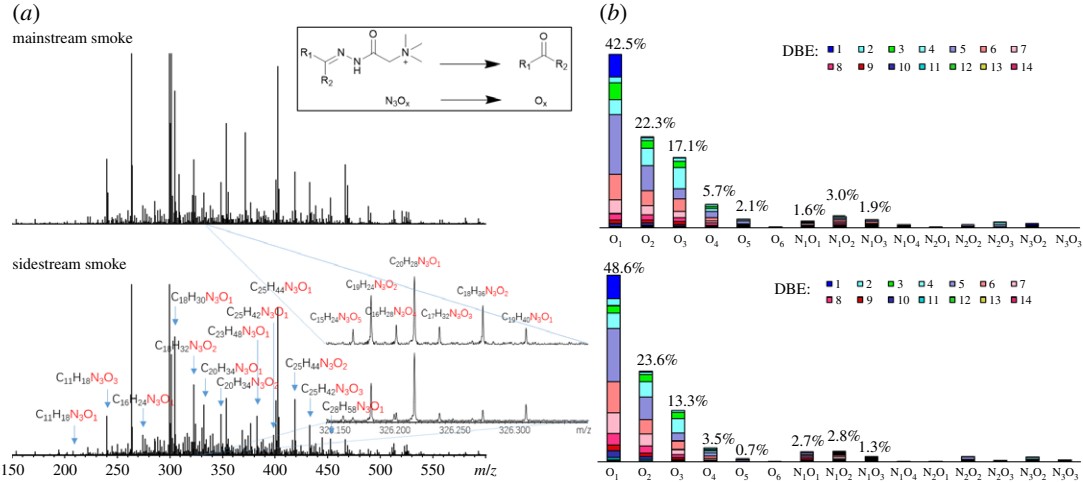

**Figure 2.** (a) Broadband +ESI FT-ICR mass spectra of the AKPs after GirT derivatization, and (b) class distribution of assigned AKPs. Note: Abundance percentage for each class is shown on the histogram.

**Table 1.** Elemental composition of the cigarette smoke and neutral fractions.

| sample | elemental compositions wt% | | | | | | H/C | content mg 20 cig$^{-1}$ |
| | C | H | O | N | S | total | | |
|---|---|---|---|---|---|---|---|---|
| MSS | 63.96 | 7.83 | 24.63 | 4.17 | 0.27 | 100.86 | 1.47 | 237.1 |
| neutral of MSS | 72.58 | 8.69 | 16.54 | 1.81 | 0.15 | 99.97 | 1.44 | 80.4 |
| SSS | 70.08 | 8.33 | 16.57 | 5.12 | 0.17 | 100.27 | 1.42 | 277.7 |
| neutral of SSS | 79.73 | 9.05 | 9.97 | 1.87 | 0.12 | 100.74 | 1.36 | 135.5 |

After removing the solvent below 10°C by a rotary evaporator, the neutral fraction was obtained. The neutral fraction and 100 mg Girard reagent were dissolved in 2 ml DCM/methanol = 1/1 (v/v) mixed solution, 0.15 ml of the catalyst acetic acid was added, and the reaction was carried out at 40°C for 5 h. After removing of the solvent by a rotary evaporator, the sample was dissolved in a mixed solvent of DCM/THF = 7/3 (v/v), and the solution was subjected to the chromatography B. Other compounds were eluted sequentially by DCM/THF = 7/3 (v/v) and DCM/methanol = 20/1 (v/v), and the pale yellow derivative remained on the silica gel. Highly polar derivatives can be eluted by DCM/methanol = 3/2 (v/v). Take a trace amount of derivatives for FT-ICR MS analysis, and the other derivatives were dissolved in 4 ml of DCM/H$_2$O = 1/1 (v/v) solvent, and 200 μl of hydrochloric acid (12 mol l$^{-1}$) was added. After reacting for 10 h at room temperature, the original carbonyl compound which was hydrolysed back was extracted into the DCM layer. The aqueous phase was extracted twice by 4 ml DCM. Then the DCM layers were combined in a flask, the solvent was removed by a rotary evaporator below 10°C. The sample was subjected to GCMS and Orbitrap MS analysis.

Finally, 20.9 and 32.9 mg of aldehydes and ketones were obtained from MSS and SSS, respectively. Table 1 shows the elemental compositions of MSS, SSS and corresponding neutral fractions.

## 2.4. Instrument conditions

The elemental analysis was performed at EL cube CHN papid OXY cube (Elementar, Germany). Carbon and hydrogen content measurement was according to the ASTM D5291-2002 method; sulfur was according to the ASTM D5453-2004; oxygen was according to the ASTM D5622-1995 and nitrogen was according to the ASTM D5762-2002.

The GCMS analyses were carried out on an Agilent 7890-5975c GCMS equipped with a HP-5 MS column (60 m × 0.25 mm × 0.25 μm). The GC oven was held at 60°C for 3 min, programmed to 120°C at a rate of 15°C min$^{-1}$, programmed to 310°C at a rate of 5°C min$^{-1}$ and then held constant at 310°C for 30 min. The injector and transfer line temperatures were held at 300°C. Helium was used as a carrier gas at a flow rate of 1 ml min$^{-1}$. The mass spectrometry (MS) was used as chromatographic

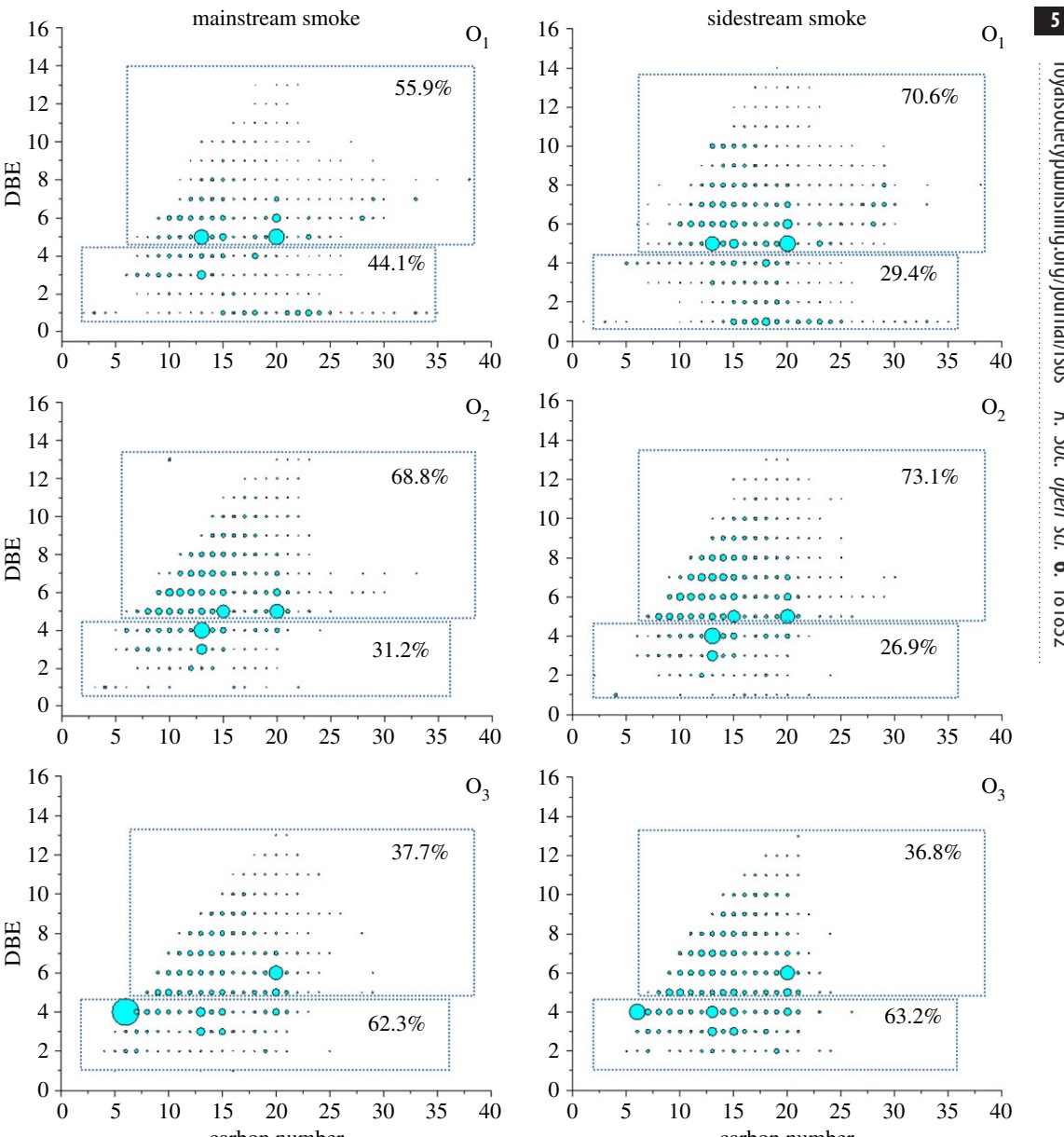

**Figure 3.** Ion relative abundance plots of DBE versus carbon number for $O_1$, $O_2$ and $O_3$ class species. The abundance percentages of the two part for each class were calculated and shown on the dot plot.

detector. The electron impact (EI) ionization source was operated under 70 eV ionization energy. The MS ion source was at 230°C. The mass range was 35–420 $m/z$ with a 0.5 s scan period.

For +ESI FT-ICR MS and Orbitrap MS analysis, The Girard reagent T (GirT) derivatives and its hydrolysates were respectively dissolved in DCM/MeOH (1:1, v/v) and DCM to 10 mg ml$^{-1}$, respectively. A total of 20 µl of the sample solution was further diluted with 1 ml of toluene/methanol (1:3, v/v) solution. Ammonium formate was added to AKs fractions to enhance ionization of AKs in positive-ion ESI Orbitrap MS analysis.

The GirT derivatives were analysed using a Bruker apex-ultra FT-ICR mass spectrometer equipped with a 9.4 T superconducting magnet. Sample solutions were infused via an Apollo II electrospray source at 180 µl h$^{-1}$ with a syringe pump. The positive-ion ESI operating conditions were as follows: The spray shield voltage was 4.5 kV. The capillary column front end voltage was 5 kV. The capillary column end voltage was 280 V. The collision cell accumulated time was 1.0 s, and the time-of-flight window was 0.9 ms. The mass range was set at $m/z$ 100−700. A total of 64 scans of 4 M words were accumulated before the Fourier transformation. Methodologies for FT-ICR MS mass calibration, data acquisition, and processing were reported elsewhere [51].

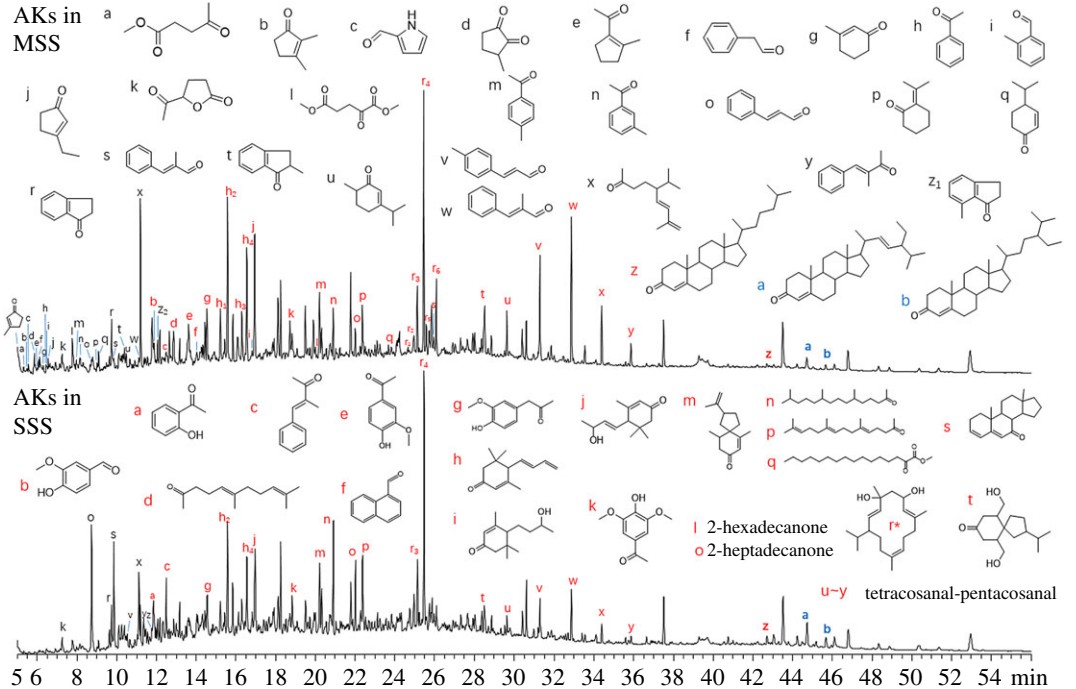

**Figure 4.** TIC of carbonyls by gas chromatography – mass spectrometry. Compound assignment was carried out through matching the mass spectra with the NIST library.

AKs fractions were analysed through a Thermo Scientific Orbitrap Fusion mass spectrometer. The positive-ion ESI operating conditions were as follows: spray voltage, 3.6 kV; sheath gas (nitrogen) flow rate, 8.0 (arbitrary units); aux gas flow rate, 3.0 (arbitrary units); ion transfer tube temperature, 280°C. The resolution was up to 500 000 at $m/z$ 200. The scanning mass range was 100–600 Da, and the spectrum was scanned for 1 min. Thermo Xcalibur Quan Browser software was used for the qualitative analysis.

## 2.5. Compounds tentative identification and quantification

Tentative identifications were based on matching the mass spectra of unknowns with those in the NIST02 (National Institute of Standards and Technology, Gaithersburg, MD, USA) mass spectral library. Quantitative data for tentatively identified compounds were obtained by the internal standard method using hexadecane as the internal standard, without considering calibration factor (i.e. $F = 1.00$ for all compounds).

# 3. Results and discussion

## 3.1. Characterization of AKPs in the MSS and SSS through selective derivatization followed by +ESI FT-ICR MS

The AKs reacted with the GirT to form a charged quaternary ammonium moiety with high conversion [28,30,37]. The quaternary ammonium moiety has strong response in positive ion mode ESI MS. The FT-ICR MS has higher resolution at high molecular weight ends compared to other MS, and it is more suitable for the analysis of GirT derivatives. Figure 2a shows a comparison of the broadband positive ion ESI FT-ICR mass spectra of GirT derivatives of the MSS and SSS. The spectra show that the molecular weight distributions are between 150 and 550 Da. Figure 2a also shows the $m/z$ scale-expanded segments at 326. The peaks with a 380 000 resolving power at $m/z = 326$. The accurate mass analysis can be used to identify the elemental compositions of these compounds. Except for a small amount of impurity peaks, substantially all of the peaks are compounds of the $N_3O_x$ type. It can be seen from the mechanism diagram that the compound corresponding to $N_3O_x$ is an $O_x$ compound. The relative abundance of each class and type (DBE) is calculated and shown in figure 2b. The AKPs

**Table 2.** Carbonyls in cigarette smoke particle phase. All compounds identified by MS database. CAS, chemical abstracts service registry number; if CAS is not available, N# (NIST number) will be given. Approx. concen., approximate concentration, assuming all response factors of 1; nd, not detected; SI, a direct matching factor for the unknown and the library spectrum; RSI, a reverse search matching factor ignoring any peaks in the unknown that are not in the library spectrum; m and s, obtained from mainstream and sidestream smoke tar, respectively. *, Determined by characteristic ions.

| $t_R$ (min) | compound | CAS | approx. concen. ($\mu$g cig$^{-1}$) | | match factor | |
|---|---|---|---|---|---|---|
| | | | MS | SS | SI | RSI |
| 5.32 | methyl levulinate | 624-45-3 | 0.11 | 0.09 | 835[m] | 882 |
| 5.49 | 2,3-dimethyl-2-cyclopenten-1-one | 1121-05-7 | 0.09 | 0.01 | 791[m] | 799 |
| 5.58 | pyrrole-2-carboxaldehyde | 1003-29-8 | 0.02 | 0.08 | 812[s] | 871 |
| 5.86 | 2-hydroxy-3-methyl-2-cyclopenten-1-one | 80-71-7 | 0.12 | 0.04 | 920[m] | 927 |
| 6.01 | 1-(2-methyl-1-cyclopentenyl)ethanone | 3168-90-9 | 0.05 | 0.02 | 744[m] | 809 |
| 6.14 | phenylacetaldehyde | 122-78-1 | 0.15 | 0.05 | 851[m] | 934 |
| 6.31 | 3-methyl-2-cyclohexen-1-one | 1193-18-6 | 0.06 | 0.03 | 728[m] | 896 |
| 6.44 | acetophenone | 98-86-2 | 0.06 | 0.04 | 898[m] | 905 |
| 6.47 | m-tolualdehyde | 620-23-5 | 0.15 | 0.16 | 922[s] | 952 |
| 6.55 | 3-ethylcyclopent-2-en-1-one | 5682-69-9 | 0.12 | 0.03 | 822[m] | 899 |
| 6.8 | 3-ethyl-2-hydroxy-2-cyclopenten-1-one | 21835-01-8 | 0.09 | 0.02 | 835[m] | 877 |
| 7.26 | 5-acetyldihydrofuran-2(3H)-one | 29393-32-6 | 0.37 | 0.52 | 919[s] | 939 |
| 7.78 | dimethyl 2-oxoglutarate | 13192-04-6 | 0.70 | 0.63 | 810[m] | 956 |
| 7.96 | 4′-methylacetophenone | 122-00-9 | 0.17 | 0.10 | 850[m] | 903 |
| 8.15 | 3′-methylacetophenone | 585-74-0 | 0.16 | 0.13 | 840[m] | 861 |
| 8.73 | trans-cinnamaldehyde | 14371-10-9 | 0.19 | 4.31 | 902[s] | 907 |
| 9.02 | 2-(1-methylethylidene)cyclohexanone | 13747-73-4 | 0.15 | 0.17 | 728[s] | 792 |
| 9.1 | 4-isopropylcyclohex-2-en-1-one | 500-02-7 | 0.44 | 0.43 | 821[m] | 845 |
| 9.74 | 1-indanone | 83-33-0 | 1.01 | 1.42 | 897[s] | 941 |
| 9.87 | alpha-methylcinnamaldehyde | 101-39-3 | 0.22 | 3.39 | 869[s] | 893 |
| 10.4 | 2-methyl-1-indanone | 17496-14-9 | 0.37 | 0.68 | 736[s] | 828 |
| 10.47 | 3-(isopropyl)-6-methylcyclohex-2-en-1-one | 499-74-1 | 0.10 | 0.09 | 822[m] | 860 |
| 10.54 | 3-(p-tolyl)acrylaldehyde | 1504-75-2 | 0.02 | 0.25 | 829[s] | 843 |
| 11.13 | alpha-methylcinnamaldehyde | 101-39-3 | 0.20 | 2.22 | 824[s] | 833 |
| 11.2 | (E)-5-isopropyl-8-methylnona-6,8-dien-2-one | 54868-48-3 | 2.51 | 0.94 | 854[m] | 861 |
| 11.36 | 3-methyl-4-phenyl-3-buten-2-one | 1901-26-4 | 0.09 | 0.76 | 763[s] | 777 |
| 11.77 | 7-methyl-1-indanone | 39627-61-7 | 0.30 | 0.76 | 814[s] | 843 |
| 11.85 | 2′-hydroxyacetophenone | 118-93-4 | nd | 2.02 | 665[s] | 861 |
| 11.89 | 3-methoxy-4-hydroxybenzaldehyde | 121-33-5 | 0.21 | 0.08 | 863[m] | 884 |
| 12.06 | 7-methyl-1-indanone | 39627-61-7 | 0.58 | 1.19 | 771[s] | 875 |
| 12.49 | 3-methyl-4-phenyl-3-buten-2-one | 1901-26-4 | 0.43 | 2.57 | 760[s] | 785 |
| 12.88 | geranyl acetone | 3796-70-1 | 0.79 | 0.81 | 709[m] | 743 |
| 13.63 | acetovanillone | 498-02-2 | 1.15 | 1.11 | 776[m] | 850 |
| 14.55 | 4-hydroxy-3-methoxyphenylacetone | 2503-46-0 | 1.36 | 2.00 | 837[s] | 865 |
| 15.21 | 4,7,9-megastigmatrien-3-one (or isomer) | 38818-55-2 | 1.05 | 1.31 | 835[s] | 859 |
| 15.58 | 4,7,9-megastigmatrien-3-one (or isomer) | 38818-55-2 | 2.89 | 3.66 | 849[s] | 872 |
| 16.3 | 4,7,9-megastigmatrien-3-one (or isomer) | 38818-55-2 | 1.36 | 1.52 | 797[s] | 883 |
| 16.55 | 4,7,9-megastigmatrien-3-one (or isomer) | 38818-55-2 | 2.18 | 2.70 | 870[s] | 892 |

(*Continued.*)

| $t_R$ (min) | compound | CAS | approx. concen. ($\mu$g cig$^{-1}$) | | match factor | |
|---|---|---|---|---|---|---|
| | | | MS | SS | SI | RSI |
| 16.82 | 4-(3-hydroxybutyl)-3,5,5-trimethylcyclohex-2-en-1-one | 36151-02-7 | 0.18 | 0.96 | 718[s] | 750 |
| 16.98 | 3,5,5-trimethyl-4-(3-hydroxy-1-butenyl)-2-cyclohexene-1-one | 34318-21-3 | 3.04 | 3.71 | 883[s] | 893 |
| 18.71 | acetosyringone | 2478-38-8 | 1.18 | 0.99 | 778[m] | 829 |
| 19.99 | hexadecan-2-one | 18787-63-8 | 0.14 | 0.32 | 749[s] | 788 |
| 20.22 | solavetivone | 54878-25-0 | 1.46 | 2.54 | 852[m] | 950 |
| 20.91 | Fitone | 502-69-2 | 0.91 | 3.67 | 893[s] | 937 |
| 22.00 | 2-heptadecanone | 2922-51-2 | 0.50 | 2.12 | 782[s] | 838 |
| 22.38 | farnesyl acetone | 1117-52-8 | 1.23 | 3.12 | 830[s] | 868 |
| 23.69 | 2-ketopalmitic acid methyl ester | 55836-30-1 | 0.27 | 0.39 | 767[s] | 889 |
| 24.65 | 12-isopropyl-1,5,9-trimethyl-4,8,13-cyclotetradecatriene-1,3-diol (or isomer) | 7220-78-2 | 0.14 | 0.43 | 689[s] | 696 |
| 24.94 | 12-isopropyl-1,5,9-trimethyl-4,8,13-cyclotetradecatriene-1,3-diol (or isomer) | 7220-78-2 | 0.46 | 1.41 | 732[s] | 741 |
| 25.13 | 12-isopropyl-1,5,9-trimethyl-4,8,13-cyclotetradecatriene-1,3-diol (or isomer) | 7220-78-2 | 1.31 | 2.39 | 845[s] | 850 |
| 25.45 | 12-isopropyl-1,5,9-trimethyl-4,8,13-cyclotetradecatriene-1,3-diol (or isomer) | 7220-78-2 | 4.56 | 8.39 | 851[s] | 854 |
| 25.57 | 12-isopropyl-1,5,9-trimethyl-4,8,13-cyclotetradecatriene-1,3-diol (or isomer) | 7220-78-2 | 0.61 | 0.44 | 840[m] | 854 |
| 25.96 | 3,5-androstadien-7-one | 32222-21-2 | 0.42 | 0.68 | 724[s] | 734 |
| 26.08 | 12-isopropyl-1,5,9-trimethyl-4,8,13-cyclotetradecatriene-1,3-diol (or isomer) | 7220-78-2 | 1.35 | 0.89 | 810[m] | 845 |
| 28.52 | 11,12-dihydroxyspirovetiva-1(10)-en-2-one | ID#: 87489 | 0.87 | 0.73 | 830[m] | 869 |
| 29.63 | henicosanal | 51227-32-8 | 0.86 | 0.62 | * | |
| 31.29 | docosanal | 57402-36-5 | 2.09 | 1.55 | * | |
| 32.88 | tricosanal | 72934-02-2 | 2.70 | 1.54 | * | |
| 34.4 | tetracosanal | 57866-08-7 | 1.15 | 0.68 | * | |
| 35.88 | pentacosanal | 58196-28-4 | 0.46 | 0.27 | * | |
| 42.71 | 5-cholesten-3-one | 601-54-7 | 0.15 | 0.53 | 833[s] | 837 |
| 44.74 | 4,22-cholestadien-3-one | 55688-43-2 | 0.39 | 1.38 | 845[s] | 887 |
| 45.69 | b-sitostenone | 1058-61-3 | 0.24 | 0.73 | 798[s] | 887 |

of the MSS and SSS include: $O_{1-6}$, $N_1O_{1-4}$, $N_2O_{1-3}$ and $N_3O_{2-3}$, corresponding to $N_3O_{1-6}$, $N_4O_{1-4}$, $N_5O_{1-3}$ and $N_6O_{2-3}$ GirT derivatives of the mass spectrum respectively. Relative abundance is defined as the magnitude of each peak divided by the sum of the magnitudes of all identified peaks (excluding the isotopic peaks) in the MS spectrum. The data shown in figure 2 are the summed mass spectrometric intensities but do not quantitatively reflect the abundance of species in the sample. The $O_1$ class AKPs has the highest relative abundance in MSS and SSS, followed by the $O_2$, $O_3$ and $O_4$ classes. Compared to SSS, the MSS has a slightly higher relative abundance of high oxygenate. This result is consistent with the elemental analysis result in table 1. This may be due to the fact that MSS is produced at higher temperatures and higher amounts of oxygen. The number of $O_{1-6}$ class AKPs and total AKPs in MSS obtained by ESI FT-ICR MS is about 740 and 1100, and in SSS is about 650 and 970. Due to the presence of isomers, the actual number will be greater.

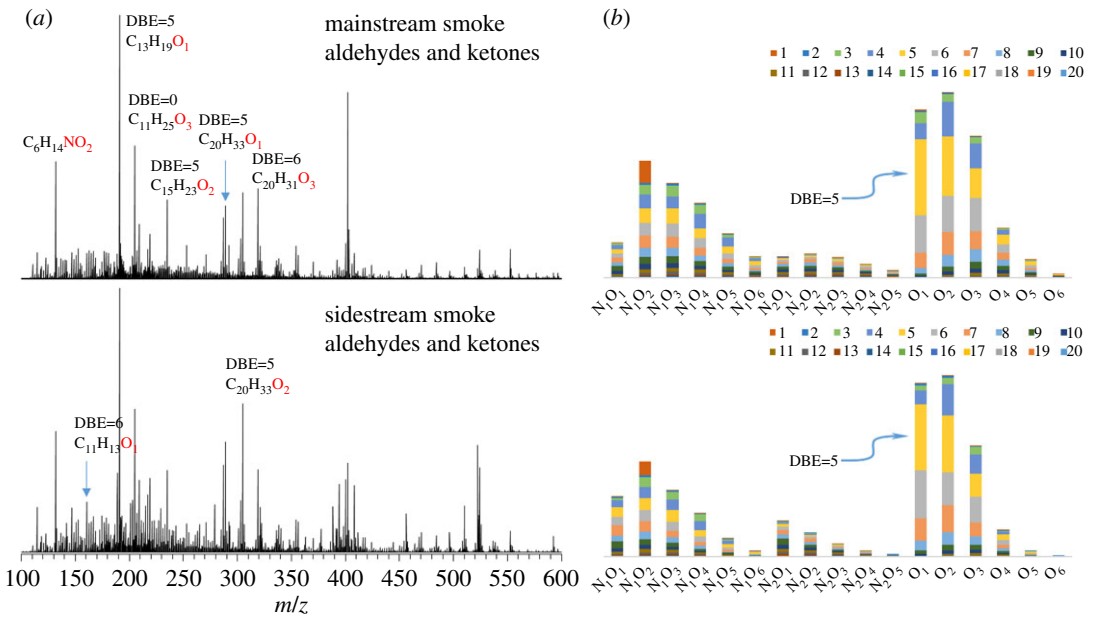

**Figure 5.** Broadband +ESI Orbitrap mass spectra of the carbonyls and the relative abundance of various assigned class species.

Figure 3 shows the *iso*-abundance plots of DBE as a function of carbon number for $O_1 \sim O_3$ species. The $O_{1-3}$ class species has a DBE value ranges (1−13) and carbon number (6−35). Since the carbonyl group contributes to an unsaturation, DBE = 1 series are saturated AKs. The AKs with DBE = 2−14 may contain unsaturated bond, naphthenic and/or aromatic ring. $O_2$ species with DBE = 1 are saturated AKs but contained a hydroxyl or an ether group. $O_2$ species with DBE > 1 may contained two carbonyl group or unsaturated bond. $O_3$ species almost did not contain DBE = 1 series, suggesting that they contained at least two carbonyl groups.

The relative abundance percentages of $O_{1-3}$ species with DBE values of 1−4 and greater than 4 for MSS and SSS were calculated, respectively. The results indicated that the SSS contain more unsaturated and condensed AKPs than MSS. This may be due to the low level of oxygen in the production of SSS facilitating the pyrolysis process. This is consistent with the H/C ratio shown in table 1.

## 3.2. Characterization of AKPs in the MSS and SSS by GCMS and +ESI Orbitrap MS

The GirT derivatives (hydrazones) can be easily separated by column chromatography and then converted to the original AKs by hydrolysis in high yield [28,29]. Figure 4 is the total ion chromatogram (TIC) of the AKPs. According to the matching of mass spectra with the NIST library, the compounds corresponding to 63 chromatographic peaks are tentatively identified. The post-treatment process may result in the loss of volatile components, so some of small carbonyl species cannot be observed in figure 4. The molecular formulae are also shown in figure 4. In addition, hundreds of peaks cannot be identified by retrieving the library.

In cigarette smoke tar, there are lots of structural analogues of the carbonyls with 5- or 6-membered ring, isoprenoids or steroid structure. Moreover, the structure of many AKs are related to isoprenoids. In the TIC of the AKPs in MSS and SSS, through matching the mass spectra with the NIST library, the highest peak and their adjacent peaks all correspond to cembrenediols (results shown in table 2). Cembrenediols are an important class of diterpenoids in tobacco. However, the cembrenediols do not have carbonyl group. They cannot react with Girard reagent or be separated into AKs fractions. Therefore, these compounds may be carbonyl compounds with similar structure to cembrenediols, which are produced from the cembrenediols in the combustion process. This will be discussed later in figure 7.

The quantitative results are listed in table 2. Figure 4 and table 2 show that the AKPs in MSS and SSS are very close, but the content is different. In addition, the content of specific carbonyl is very different to each other. The content of high-carbon linear aldehydes (henicosanal to pentacosanal) in MSS is higher than that of SSS, but the content of some low-carbon aromatic-contained aldehydes (*trans*-cinnamaldehyde, *alpha*-methylcinnamaldehyde) in sidestream smoke is higher than that of

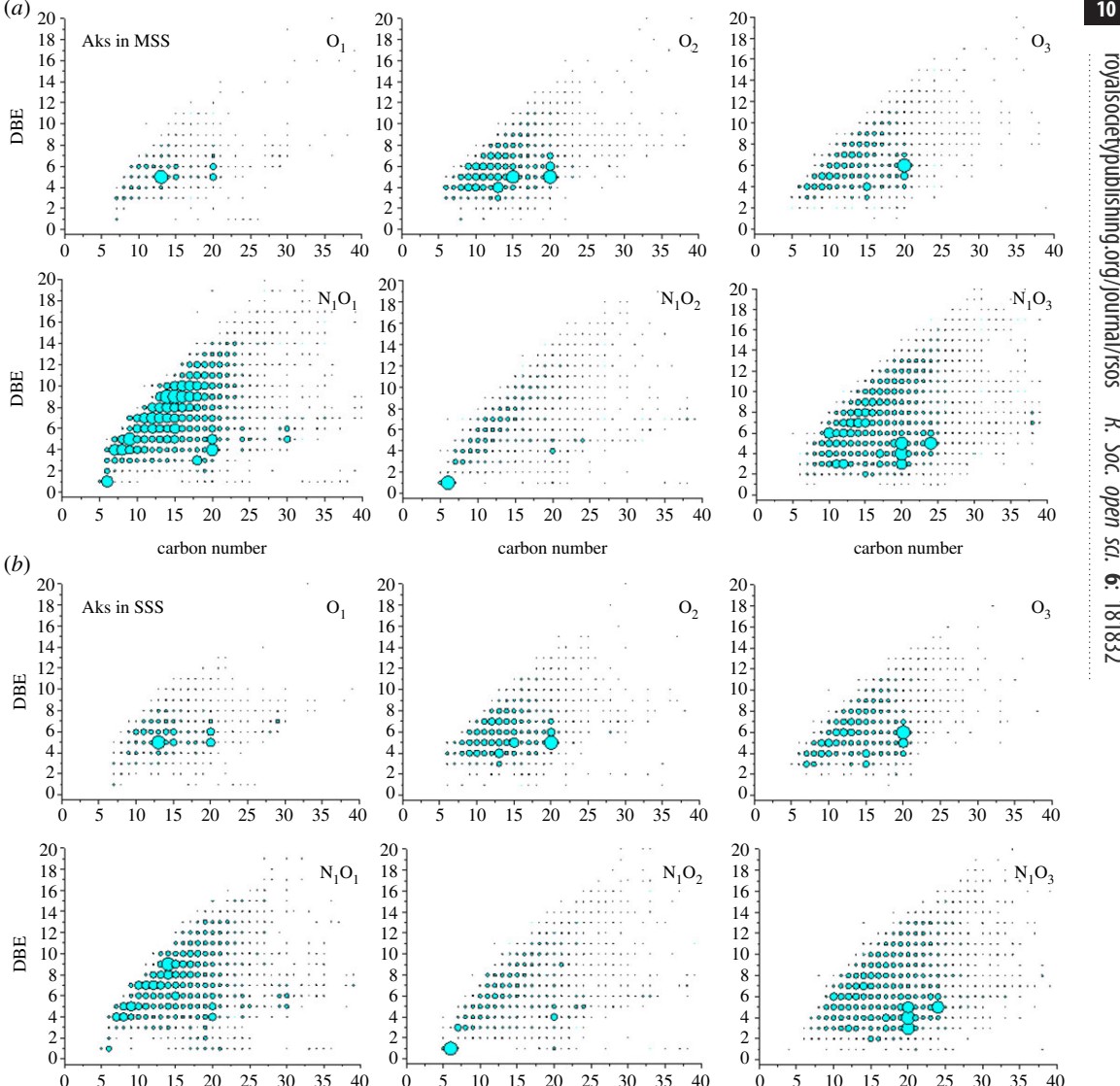

**Figure 6.** (a,b) Ion relative abundance plots of DBE versus carbon number for $O_{1-3}$ and $N_1O_{1-3}$ class species.

mainstream smoke. This may be due to the fact that MSS and SSS are formed in different ways of combustion. The higher temperatures and higher amounts of oxygen lead to the breaking of the carbon chain to form more linear aldehydes during MSS production. The low level of oxygen in the production of SSS facilitates the pyrolysis process and forms more unsaturated and condensed AKPs.

As shown in figure 4a,b, a large amount of component cannot be resolved by the GC. To obtain a comprehensive result, the high resolution mass spectrometry is used to analyse the sample. Comparing with the FT-ICR MS, the Orbitrap MS has better ion transmission both in the lower and higher molecular weight region. It is considered that the FT-ICR MS has higher resolution at high molecular weight region. Since the molecular weight of original AKPs is much lower than the corresponding GirT derivatives, the Orbitrap MS was selected to analyse the original AKPs.

Figure 5a shows the mass spectra by positive ion ESI Orbitrap MS. The average molecular weight of figure 5a moved forward (212 Da) compared to the figure 2a (348 Da), which is due to the molecular weight of the AKs being smaller than the corresponding hydrazine derivatives. Figure 5b also shows that the relative abundance of $N_xO_y$ and high-oxygen numbered compounds is much higher than in figure 2b. This is because the ionized functional groups of GirT derivatives are all quaternary ammonium groups. After hydrolysing the hydrazine groups, the $N_xO_y$ and high-oxygen numbered AKPs exhibit higher ionization efficiencies than others. This makes it possible to analyse the $N_xO_y$ and high-oxygen numbered AKPs more clearly.

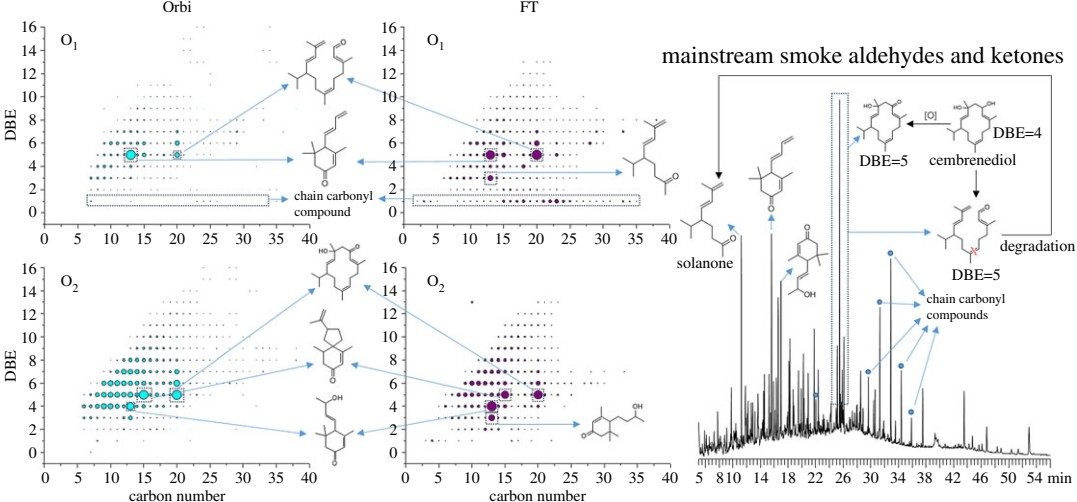

**Figure 7.** Comparison of Orbitrap MS and FT-ICR MS results and correspondence between high-abundance mass spectral peaks and chromatographic peaks.

Figure 6 shows the ion relative abundance plots of DBE versus carbon number for $O_{1-3}$ and $N_1O_{1-3}$ class species of the smoke tar by positive-ion ESI Orbitrap MS. The DBE values of $O_{1-3}$ carbonyls are concentrated at 3–10. $N_1O_{1-3}$ carbonyls are more condensed and unsaturated than $O_{1-3}$ carbonyls.

Figure 7 shows the ion relative abundance plots of DBE versus carbon number for $O_1$ and $O_2$ class species of the AKs in MSS by Orbitrap MS and the GirT derivatives of AKs in MSS by FT-ICR MS, and the TIC of AKs in MSS by GCMS. At the carbon number of 20, the $O_1$ and $O_2$ compounds with DBE = 5 have higher abundance. By aligning with the identified compounds in GCMS, they may be the AKs produced by the degradation of cembrenediol during combustion. The cembrenediol [$O_2$, DBE = 4] may be oxidized to an alcohol-ketone [$O_2$, DBE = 5] during combustion or converted to a monoaldehyde [$O_1$, DBE = 5] by chain scission and dehydration. Higher levels of solanone prove the possibility of this process. Compared with the FT-ICR MS results, the abundance of low DBE aldehydes in Orbitrap MS is lower. The difference between the Orbitrap and the FT-ICR is probably due to the different ionization efficiency between GirT derivatives of AKs and AKs itself, as discussed previously in figure 5.

## 4. Conclusion

In this work, the AKPs of MSS and SSS were successfully separated and analysed in detail. A total of 63 AKs were identified and quantified by GCMS. Due to the complex post-treatment of chemical derivatization, a portion of volatile or unstable AKs are inevitably lost, so the number of AKs that were characterized by GCMS is less than already reported [52]. Fifteen species of AKs were characterized by ESI FT-ICR MS and Orbitrap MS: $O_{1-6}$, $N_1O_{1-4}$, $N_2O_{1-3}$ and $N_3O_{2-3}$. Regardless of the isomer, the total number of aldehydes and ketones obtained by ESI FT-ICR MS in MSS and SSS is about 1100 and 970, respectively. This number is much higher than the reported [52], which indicated a large amount of component cannot be resolved by the GCMS. Furthermore, The nitrogen-containing carbonyls are more condensed and unsaturated than $O_{1-3}$ carbonyls. The AKPs in MSS are more oxygenated and less unsaturated than that in SSS. This is consistent with the previous study [4]. Future work should focus on the toxicity of the isolated carbonyls to humans and the environment.

Data accessibility. Our raw data (GCMS, Orbitrap MS and FT-ICR MS) are deposited at Dryad: http://dx.doi.org/10.5061/dryad.6v1r5n8 [53].

Authors' contributions. X.C. participated in the laboratory work and data analysis; Q.S. participated in the design of the study and helped to revise the paper; X.Z. carried out the design of the study, participated in the laboratory work, participated in data analysis and wrote the paper; X.L. participated in the design of the study and helped to revise the paper. All authors gave final approval for publication.

Competing interests. We declare we have no competing interests.

Funding. This work was supported by the National Natural Science Foundation of China (NSFC, 21405069), Excellent Talents programme of Liaoning Provincial Universities (Department of Education of Liaoning Province) (LJQ2015068), Natural Science Foundation of Liaoning Province (201602339).

Acknowledgements. The authors thank all reviewers for valuable comments.

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
