## [Reviewer comments · Royal Society Open Science]

Review History

RSOS-181832.R0 (Original submission)

Review form: Reviewer 1

Is the manuscript scientifically sound in its present form?

Yes

Are the interpretations and conclusions justified by the results?

Yes

Is the language acceptable?

No

Is it clear how to access all supporting data?

Yes

Do you have any ethical concerns with this paper?

No

Have you any concerns about statistical analyses in this paper?

No

Recommendation?

Major revision is needed (please make suggestions in comments)

Comments to the Author(s)

Major revision: mainly with respect to the quantifications, see details in the enclosed review (Appendix A).

Review form: Reviewer 2

Is the manuscript scientifically sound in its present form?

No

Are the interpretations and conclusions justified by the results?

No

Is the language acceptable?

Yes

Is it clear how to access all supporting data?

No

Do you have any ethical concerns with this paper?

No

Have you any concerns about statistical analyses in this paper?

I do not feel qualified to assess the statistics

Recommendation?

Major revision is needed (please make suggestions in comments)

Comments to the Author(s)

This paper presents an interesting piece of work dealing with the investigation of aldehydes and ketones (AKs) in complex mixture by mass spectrometry using a derivatization step by the Girard's T reagent. This methodology is used to compare the AK composition of two different cigarette smoke the MSS and the SSS. In spite of the interest of this work significant major revision are required for different reasons.

The introduction requires to be improved and modified. It has to be more clearly focused on the analysis of cigarette smoke by non-targeted mass spectrometry. Significant references are missing as those of the Zimmerman group and those of other groups. In the last five years, a number of papers reported the analysis of AKs in complex mixture by HR MS after a reduction or a derivitization step. The papers have to be cited and discussed in the introduction (and compared to the proposed methodology in the conclusion).

One important issue in this paper is the use of two HR MS instruments. The authors justified the use of orbitrap according to the low mass cut-off of FT ICRMS which is too high to observe the "naked" AKs. This is correct, but the orbitrap instrument has the capability to investigate both the low mass and the high mass range. Moreover, authors observed on the orbitrap mass spectra

ion at higher m/z than on the FT ICR MS, this means that a part of the AKs, which are derivatized by Girard's T reagent are not detected. Consequently, the assertion of the authors concerning the same ionization efficiency for all compounds after reaction of AKs with Girard's reactant is totally wrong. The comparison of the DBE distribution for a given class compound (O1 for example) is significantly different why? This deserves to be deeper discuss. What about the compounds which contain two or more carbonyl chemical groups?

One other important issue is relative to the detection N_xO_y compounds by ESI-FT ICR MS: how are the author sure that these species are associated with derivatized MSS/or SSS compounds. A large part of cigarette smoke simultaneously contains nitrogen and oxygen see the paper corresponding to ref 4 and the other papers of the same group. In the same way how are the author sure that only AKs are present in the so-called AKs fraction? Without this confirmation the obtained conclusions are highly disputable.

What about the cotinine behavior which is a well-known cigarette smoke tracer?

Why the authors used paper filters? More adequate filters are quartz filter for which it can be considered that it cannot interact with the smoke sample.

Why collect the sample for 20 cigarettes in the same time. Interaction and chemical reaction may happen between the still collected particles and the compounds (in gas and PM phase). This may have a significant importance after the 20 smoking procedure.

Details of the extraction step have to be given.

The non-controlled evaporation of the solvent may lead to the evaporation of the more volatile cigarette smoke components.

The mass resolution of FT ICR MS measurement has to be given.

Details of the "tentative identification and quantification" page 3 line 57/60. The quantification aspect is not discussed in the results and discussion section.

The figures are too small and very hard to be clearly examined. What's mean MK and CK in Fig 6.

Finally, authors have to give a comparison between what their findings are and what it is reported in the literature.

Decision letter (RSOS-181832.R0)

27-Nov-2018

Dear Dr Zhou:

Title: Molecular characterization of aldehydes and ketones in particle phase of mainstream and sidestream cigarette smoke

Manuscript ID: RSOS-181832

The editor assigned to your manuscript has now received comments from reviewers. We would like you to revise your paper in accordance with the referee and Subject Editor suggestions which can be found below (not including confidential reports to the Editor). Please note this decision does not guarantee eventual acceptance.

Please submit your revised paper before 20-Dec-2018. Please note that the revision deadline will expire at 00.00am on this date. If we do not hear from you within this time then it will be assumed that the paper has been withdrawn. In exceptional circumstances, extensions may be

possible if agreed with the Editorial Office in advance. We do not allow multiple rounds of revision so we urge you to make every effort to fully address all of the comments at this stage. If deemed necessary by the Editors, your manuscript will be sent back to one or more of the original reviewers for assessment. If the original reviewers are not available we may invite new reviewers.

Please also include the following statements alongside the other end statements. As we cannot publish your manuscript without these end statements included, if you feel that a given heading is not relevant to your paper, please nevertheless include the heading and explicitly state that it is not relevant to your work.

- Ethics statement

Please clarify whether you received ethical approval from a local ethics committee to carry out your study. If so please include details of this, including the name of the committee that gave consent in a Research Ethics section after your main text. Please also clarify whether you received informed consent for the participants to participate in the study and state this in your Research Ethics section.

OR

Please clarify whether you obtained the necessary licences and approvals from your institutional animal ethics committee before conducting your research. Please provide details of these licences and approvals in an Animal Ethics section after your main text.

OR

Please clarify whether you obtained the appropriate permissions and licences to conduct the fieldwork detailed in your study. Please provide details of these in your methods section.

RSC Associate Editor:
Comments to the Author:
(There are no comments.)

RSC Subject Editor:
Comments to the Author:
(There are no comments.)

Reviewers' Comments to Author:
Reviewer: 1

Comments to the Author(s)
Major revision: mainly with respect to the quantifications, see details in the enclosed review.

Reviewer: 2

Comments to the Author(s)

This paper presents an interesting piece of work dealing with the investigation of aldehydes and ketones (AKs) in complex mixture by mass spectrometry using a derivatization step by the Girard's T reagent. This methodology is used to compare the AK composition of two different cigarette smoke the MSS and the SSS. In spite of the interest of this work significant major revision are required for different reasons.

The introduction requires to be improved and modified. It has to be more clearly focused on the analysis of cigarette smoke by non-targeted mass spectrometry. Significant references are missing as those of the Zimmerman group and those of other groups. In the last five years, a number of papers reported the analysis of AKs in complex mixture by HR MS after a reduction or a derivitization step. The papers have to be cited and discussed in the introduction (and compared to the proposed methodology in the conclusion).

One important issue in this paper is the use of two HR MS instruments. The authors justified the use of orbitrap according to the low mass cut-off of FT ICRMS which is too high to observe the "naked" AKs. This is correct, but the orbitrap instrument has the capability to investigate both the low mass and the high mass range. Moreover, authors observed on the orbitrap mass spectra ion at higher m/z than on the FT ICR MS, this means that a part of the AKs, which are derivatized by Girard's T reagent are not detected. Consequently, the assertion of the authors concerning the same ionization efficiency for all compounds after reaction of AKs with Girard's reactant is totally wrong. The comparison of the DBE distribution for a given class compound (O1 for example) is significantly different why? This deserves to be deeper discuss. What about the compounds which contain two or more carbonyl chemical groups?

One other important issue is relative to the detection NxOy compounds by ESI-FT ICR MS: how are the author sure that these species are associated with derivatized MSS/or SSS compounds. A large part of cigarette smoke simultaneously contains nitrogen and oxygen see the paper corresponding to ref 4 and the other papers of the same group. In the same way how are the author sure that only AKs are present in the so-called AKs fraction? Without this confirmation the obtained conclusions are highly disputable.

What about the cotinine behavior which is a well-known cigarette smoke tracer?

Why the authors used paper filters? More adequate filters are quartz filter for which it can be considered that it cannot interact with the smoke sample.

Why collect the sample for 20 cigarettes in the same time. Interaction and chemical reaction may happen between the still collected particles and the compounds (in gas and PM phase). This may have a significant importance after the 20 smoking procedure.

Details of the extraction step have to be given.

The non-controlled evaporation of the solvent may lead to the evaporation of the more volatile cigarette smoke components.

The mass resolution of FT ICR MS measurement has to be given.

Details of the “tentative identification and quantification” page 3 line 57/60. The quantification aspect is not discussed in the results and discussion section.

The figures are too small and very hard to be clearly examined. What’s mean MK and CK in Fig 6.

Finally, authors have to give a comparison between what their findings are and what it is reported in the literature.

Author's Response to Decision Letter for (RSOS-181832.R0)

See Appendix B.

Decision letter (RSOS-181832.R1)

14-Jan-2019

Dear Dr Zhou:

Title: Molecular characterization of aldehydes and ketones in particle phase of mainstream and sidestream cigarette smoke

Manuscript ID: RSOS-181832.R1

It is a pleasure to accept your manuscript in its current form for publication in Royal Society Open Science. The chemistry content of Royal Society Open Science is published in collaboration with the Royal Society of Chemistry.

Yours sincerely,

Dr Laura Smith

Publishing Editor, Journals

Royal Society of Chemistry

Thomas Graham House

Science Park, Milton Road

Cambridge, CB4 0WF

Royal Society Open Science - Chemistry Editorial Office

RSC Associate Editor
Comments to the Author:
(There are no comments.)

Reviewer(s)' Comments to Author:

Appendix A

RSOS-181832 - review

In this manuscript the authors use a well-established technique for the, mainly, qualitative analysis of carbonyl compounds in cigarette smoke. They demonstrate that a good amount of quantitative data can be gained this way on a complex mixture of related compounds. This gives the work the value that warrants a publication of it.

The text is written in good English but it should be polished, in some places it is somewhat bumpy or simply incorrect – one example (line 38 on page 2): “After reacted with the Girard T reagent, the polarity of carbonyls produced quarternary ammonium salt derivatives is much higher than other components in the neutral fraction and is easily separated.” It would be a pity not to correct such expressions.

Some further questions that should be cleared up are:

- Although the expression “Girard’s reagent” is known, it is my sense that “Girard reagent” is much more common

- What detector was used in gas chromatography? In the section “Instrument conditions” I don’t find any explicit mentioning of this.

- Why did the authors used both Orbitrap and FT-ICR MS? In the section mentioned above I don’t see any rationale for this. If the reason was the better ion transmission for Orbitrap in the lower molecular weight region, why then use FT-ICR at all? And is this assumption borne out by the results (e.g. Figure 7)?

- Quantification.

This is a mine field and it is simply impossible to deduce quantitative relationships from the signal heights for two compounds, as is done in GC. Despite this, the authors use a quantitation for the Girard derivatives by postulating, without further justification, that since they all have the same permanent ionic structure, their “ionization efficiens are basically the same” (line 31, page 4). Is this assumption really true? Do the authors not neglect the hydrophobicity effect, described by John Fenn himself, that is known to have a major effect on the signal height in ESI-MS?

This effect has been described in the literature for compounds of a very similar kind to that used here, namely in ref. 18 and also in DOI 10.1021/ef5028108 It was shown that compounds present in equimolar amounts easily gave MS signals that differed by a factor of 2.

There are also uncertainties growing out of the data presented by the authors. Take as an example figure 7. In the FT O1 Kendrick plot, the two largest dots are of approximately the same size. However, in the gas chromatogram, that of the open structure is ca twice as large as that of the cyclohexeneone. In view of this, how do the authors justify quantifying based only on MS signal heights?

Furthermore, how do the authors justify the abundance percentages for different compound classes (Figure 2, Figure 5) when knowing that N containing compounds usually have a much higher ionization efficiency in ESI than other compounds, thus producing considerably higher signal heights for the same molar amounts?

The authors state on line 6, page 2: “The pre-charged property of Girard’s reagent can enhance and uniform ionization efficiency of the derivatives in ESI-MS analysis.[36]”. I certainly agree with the enhancement but the quoted sentence is not supported by the reference indicated; this reference is mainly an overview of different reagents that have been used for LC-MS, including carbonyl compounds.

- Lines 33-35 on page 2: The first two sentences in this paragraph seem to be identical.

- Table 1: Do the standard deviations justify four significant figures in the percentage numbers?

- Table 2: The first letter in the compound names should be either capitalized or not capitalized throughout, not mixed

Misspelling of “7-methyl-1-indanone”

“4,7,9-Megastigmatrien-3-one” – obviously there are several isomers, thus it would be appropriate to indicate this here and on each line add “or isomer”. The same for “12-Isopropyl-1,5,9-trimethyl-4,8,13-cyclotetradecatriene-1,3-diol”

The compound at 16.98 minutes: There are two names here, please separate them, perhaps by putting the long name in parenthesis. Please don’t use capital C in cyclohexen.

The compound at 5.86 min: The correct name according to CAS is 2-hydroxy-3-methyl-2-cyclopenten-1-one.

Figure 1: In the first “round-bottom flask”, the amounts of the different compounds is given but not that of the Girard reagent. Is that intentional?

Appendix B

We would like to thank the reviewer for providing constructive comments and help in improving the contents of this manuscript. We have revised the manuscript according to the reviewer's suggestions. We have responded to each comment in detail as shown below. Our responses are formatted in italics.

RSOS-181832 – review1

In this manuscript the authors use a well-established technique for the, mainly, qualitative analysis of carbonyl compounds in cigarette smoke. They demonstrate that a good amount of quantitative data can be gained this way on a complex mixture of related compounds. This gives the work the value that warrants a publication of it.

We thank the reviewer for these comments.

The text is written in good English but it should be polished, in some places it is somewhat bumpy or simply incorrect – one example (line 38 on page 2): “After reacted with the Girard T reagent, the polarity of carbonyls produced quarternary ammonium salt derivatives is much higher than other components in the neutral fraction and is easily separated.” It would be a pity not to correct such expressions.

Response: Thanks very much for your comments, which are very helpful for us to improve the manuscript, and our language should be improved. After carefully check, we found many grammar and sentence errors, and have modified the manuscript. Furthermore, we have invited several English teachers help correct grammar and sentences, and we

hope the revised paper will be more clear and accurate on expressions.

For example, the sentence have been revised to: "After reacted with the Girard's reagent T, the resulting aldehyde or ketone derivative is very easy to separate. This is because the polarity of the quaternary ammonium salt is much higher than the other components in the neutral fraction."

Some further questions that should be cleared up are:

- Although the expression "Girard's reagent" is known, it is my sense that "Girard reagent" is much more common

Response: Thanks for the comment. We have changed 'Girard's reagent' to 'Girard reagent' (but we retained the term "Girard's reagent T" according to other literature).

- What detector was used in gas chromatography? In the section "Instrument conditions" I don't find any explicit mentioning of this.

Response: We used mass spectrometry as the gas chromatographic detector. We have add the following sentence in the revised manuscript:

"The mass spectrometry (MS) was used as chromatographic detector. The electron impact (EI) ionization source was operated

under 70 eV ionization energy. The MS ion source was at 230 °C. The mass range was 35-420 m/z with a 0.5 s scan period.”

- Why did the authors used both Orbitrap and FT-ICR MS? In the section mentioned above I don't see any rationale for this. If the reason was the better ion transmission for Orbitrap in the lower molecular weight region, why then use FT-ICR at all? And is this assumption borne out by the results (e.g. Figure 7)?

Response: The Orbitrap MS has better ion transmission both in the lower and higher molecular weight region, and it is suitable for the analysis of original AKPs. The FT-ICR MS has higher resolution at high molecular weight ends (i.e. at 400 m/z) compared to Orbitrap MS, and it is more suitable for the analysis of GirT derivatives.

In fact, we found that for cigarette samples, the resolution of the Orbitrap MS on the high molecular weight ends is sufficient. But we want to compare these two results, so we finally chose to list the results obtained by the two mass spectrometers. We have added the following sentence in the revised manuscript:

“To obtain a comprehensive result, the high resolution mass spectrometry is used to analyze the sample. Comparing with the FT-ICR MS, The Orbitrap MS has better ion transmission both in the lower

and higher molecular weight region. It's considered that the FT-ICR MS has higher resolution at high molecular weight region. Since the molecular weight of original AKPs is much lower than the corresponding GirT derivatives, the Orbitrap MS was selected to analyse the original AKPs."

- Quantification.

This is a mine field and it is simply impossible to deduce quantitative relationships from the signal heights for two compounds, as is done in GC. Despite this, the authors use a quantitation for the Girard derivatives by postulating, without further justification, that since they all have the same permanent ionic structure, their "ionization efficiens are basically the same" (line 31, page 4). Is this assumption really true? Do the authors not neglect the hydrophobicity effect, described by John Fenn himself, that is known to have a major effect on the signal height in ESI-MS?

This effect has been described in the literature for compounds of a very similar kind to that used here, namely in ref. 18 and also in DOI 10.1021/ef5028108 It was shown that compounds present in equimolar amounts easily gave MS signals that differed by a factor of 2.

There are also uncertainties growing out of the data presented by the authors. Take as an example figure 7. In the FT O1 Kendrick plot, the two largest dots are of approximately the same size. However, in the gas chromatogram, that of the open structure is ca twice as large as that of the cyclohexeneone. In view of this, how do the authors justify quantifying based only on MS signal heights?

Response: We are sorry that we have made a mistake. We did not fully considered the hydrophobicity effect to ionization efficiency. Our expression "Since the highly conversion of AKs reaction with Girard's reagent, and the ionized groups after derivatization are all quaternary ammonium groups (the ionization efficiencies are basically the same)"

is totally wrong. We have removed this sentence in the revised manuscript. The related content has also been modified accordingly. Furthermore, we have removed and revised other statements about 'justify quantifying based only on MS signal heights' in the revised manuscript.

Furthermore, how do the authors justify the abundance percentages for different compound classes (Figure 2, Figure 5) when knowing that N containing compounds usually have a much higher ionization efficiency in ESI than other compounds, thus producing considerably higher signal heights for the same molar amounts?

Response: We are sorry that we have caused a misunderstanding.

The data shown in Figure 2 and Figure 5 are the summed mass spectrometric intensities but do not quantitatively reflect the abundance of species in the sample. We have added the following sentence in the revised manuscript:

"The relative abundance of each class and type (DBE) is calculated and shown in Figure 2(b). Relative abundance is defined as the magnitude of each peak divided by the sum of the magnitudes of all identified peaks (exclude the isotopic peaks) in the MS spectrum. The data shown in Figure 2 are the summed mass spectrometric intensities but do not quantitatively reflect the abundance of species in

the sample. The O₁ class AKPs with the most relative abundance in MSS and SSS, followed by the O₂, O₃, and O₄ class. Compared to SSS, the MSS with a little higher relative abundance of high oxygenate. This result is consistent with the elemental analysis result in Table 1.”

The authors state on line 6, page 2: “The pre-charged property of Girard’s reagent can enhance and uniform ionization efficiency of the derivatives in ESI-MS analysis.[36]”. I certainly agree with the enhancement but the quoted sentence is not supported by the reference indicated; this reference is mainly an overview of different reagents that have been used for LC-MS, including carbonyl compounds.

Response: According to your suggestion, we have revised the text:

“The pre-charged property of Girard’s reagent can enhance the ionization efficiency of the derivatives in ESI-MS analysis.[36]”

- Lines 33-35 on page 2: The first two sentences in this paragraph seem to be identical.

Response: According to your suggestion, we have revised the text.

- Table 1: Do the standard deviations justify four significant figures in the percentage numbers?

Response: The elemental analysis was performed at EL cube CHN and rapid OXY cube (Elementar, Germany) by a dedicated laboratory technician at the State Key Laboratory of Heavy Oil Processing, China University of Petroleum (Beijing). The accuracy and repeatability of

elemental analysis is recognized to be very high, and the general results of elemental analysis are four significant figures. In this study, the elemental analysis result is the average of three measurements, and the standard deviations justify four significant figures.

We have added the following sentence in the revised manuscript:

“The elemental analysis was performed at EL cube CHN rapid OXY cube (Elementar, Germany). Carbon and hydrogen content measurement was according to the ASTM D5291-2002 method; sulfur was according to the ASTM D5453-2004; oxygen was according to the ASTM D5622-1995 and nitrogen was according to the ASTM D5762-2002.”

- Table 2: The first letter in the compound names should be either capitalized or not capitalized throughout, not mixed Misspelling of “7-methyl-1-indanone”

“4,7,9-Megastigmatrien-3-one” – obviously there are several isomers, thus it would be appropriate to indicate this here and on each line add “or isomer”. The same for “12- Isopropyl-1,5,9-trimethyl-4,8,13-cyclotetradecatriene-1,3-diol”

The compound at 16.98 minutes: There are two names here, please separate them, perhaps by putting the long name in parenthesis. Please don't use capital C in cyclohexen.

The compound at 5.86 min: The correct name according to CAS is 2-hydroxy-3-methyl-2-cyclopenten-1-one.

Response: According to your suggestion, we have revised the text.

Figure 1: In the first “round-bottom flask”, the amounts of the different compounds is given but not that of the Girard reagent. Is that intentional?

Response: It is our negligence that the amount of Girard reagent is not given in Figure 1. The amount of Girard reagent is given in the “Materials and Methods section-Derivatization and separation of aldehydes and ketones”, and we also add this data in Figure 1 in the revised manuscript.

We would like to thank the reviewer for providing constructive comments and help in improving the contents of this manuscript. We have revised the manuscript according to the reviewer's suggestions. We have responded to each comment in detail as shown below. Our responses are formatted in italics.

Comments to the Author(s)-2

This paper presents an interesting piece of work dealing with the investigation of aldehydes and ketones (AKs) in complex mixture by mass spectrometry using a derivatization step by the Girard's T reagent. This methodology is used to compare the AK composition of two different cigarette smoke the MSS and the SSS. In spite of the interest of this work significant major revision are required for different reasons.

The introduction requires to be improved and modified. It has to be more clearly focused on the analysis of cigarette smoke by non-targeted mass spectrometry. Significant references are missing as those of the Zimmerman group and those of other groups. In the last five years, a number of papers reported the analysis of AKs in complex mixture by HR MS after a reduction or a derivitization step. The papers have to be cited and discussed in the introduction (and compared to the proposed methodology in the conclusion).

Response: Thanks for your suggestion. According to your suggestion, we have added the following text in the introduction.

"The AKs in the complex matrix can be directly analyzed by gas chromatography-mass spectrometry (GCMS)[11, 12] or other technique. Recently, Zimmermann and other groups[13-22] develop a series of rapid and sensitive non-targeted mass spectrometry

technique to comprehensive analysis the carbonyls and other components in complex matrix. The non-targeted method has the advantage that there are no complicated post-processing steps, fast and accurate.

One important issue in this paper is the use of two HR MS instruments. The authors justified the use of orbitrap according to the low mass cut-off of FT ICRMS which is too high to observe the “naked” AKs. This is correct, but the orbitrap instrument has the capability to investigate both the low mass and the high mass range. Moreover, authors observed on the orbitrap mass spectra ion at higher m/z than on the FT ICR MS, this means that a part of the AKs, which are derivatized by Girard’s T reagent are not detected. **Consequently, the assertion of the authors concerning the same ionization efficiency for all compounds after reaction of AKs with Girard’s reactant is totally wrong.**

Response: According to your suggestion and the comment from another reviewer, our expression “Since the highly conversion of AKs reaction with Girard’s reagent, and the ionized groups after derivatization are all quaternary ammonium groups (the ionization efficiencies are basically the same)” is totally wrong. We have removed this sentence in the revised manuscript. The related content has also been modified accordingly. Furthermore, we have removed and revised other statements about ‘justify quantifying based only on MS signal heights’ in the revised manuscript.

The comparison of the DBE distribution for a given class compound (O1 for

example) is significantly different why? This deserves to be deeper discuss. What about the compounds which contain two or more carbonyl chemical groups?

Response: According to your suggestion, the following text were revised:

“Figure 3 shows the iso-abundance plots of DBE as a function of carbon number for O1 ~ O3 species. The O1 and O3 class species has a DBE value ranges (1 – 13) and carbon number (6-35). Since the carbonyl group contributes to an unsaturation, DBE = 1 series are saturated AKs. The AKs with DBE=2-14 may contain unsaturated bond, naphthenic and/or aromatic ring. O2 species with DBE = 1 are saturated AKs but contained a hydroxyl or an ether group. O2 species with DBE > 1 may contained two carbonyl group or unsaturated bond. O3 species almost not contained DBE=1 series, suggest that at least contained two carbonyl groups.

The abundance percentages of O1-3 species with DBE values of 1-4 and greater than 4 for MSS and SSS were calculated, respectively. The results indicated that the SSS contain more unsaturated and condensed AKPs than MSS. This may be due to the low level of oxygen

in the production of SSS facilitates the pyrolysis process. This consists with the H/C ratio shown in Table 1.”

One other important issue is relative to the detection N_xO_y compounds by ESI-FT ICR MS: how are the author sure that these species are associated with derivatized MSS/or SSS compounds. A large part of cigarette smoke simultaneously contains nitrogen and oxygen see the paper corresponding to ref 4 and the other papers of the same group. In the same way how are the author sure that only AKs are present in the so-called AKs fraction? Without this confirmation the obtained conclusions are highly disputable.

Response: The original AKs are neutral compound, so first step the high-polarity acidic and basic fractions were removed by HCl and KOH modified silica gel (solid phase extraction method). In this step, the dichloromethane (DCM)/tetrahydrofuran (THF) = 7/3 (v/v) mixture was used as the eluent to elute the neutral fraction. The polarity of DCM/THF = 7/3 (v/v) is high enough to elute the aldehydes and ketones.

In the second step, the neutral fraction mixed with Girard reagent and the AKs are converted into corresponding GirT derivatives. The resulting GirT derivatives are very easy to separate. This is because the polarity of the quaternary ammonium salt is much higher than the other components in the neutral fraction. The mixture was subjected to the chromatography and the DCM / methanol = 20/1 (v/v) was used as the eluent. Because the polarity of DCM / methanol = 20/1 (v/v) is much

higher than the DCM/THF = 7/3 (v/v), the other neutral compounds were removed very cleanly. The GirT derivatives still retain on the silica gel and cannot be eluted due to its high polarity. Then a much higher polarity eluent, DCM / methanol = 3/2 was used to elute the GirT derivatives. The huge difference in polarity between these two parts ensure only AKs are present in the AKs fraction. This can be ensured by the methodological experiments in another article (10.1021/acs.energyfuels.7b03630).

We have added the following sentence in the revised manuscript:

“The original AKs are neutral compound, so first step the high-polarity acidic and basic fractions were removed by HCl and KOH modified silica gel (solid phase extraction method). In this step, the dichloromethane (DCM)/tetrahydrofuran (THF) = 7/3 (v/v) mixture was used as the eluent to elute the neutral fraction. The polarity of DCM/THF = 7/3 (v/v) is high enough to elute the aldehydes and ketones.

In the second step, the neutral fraction mixed with Girard reagent and the Aks are converted into corresponding GirT derivatives. The resulting GirT derivatives are very easy to separate. This is because

the polarity of the quaternary ammonium salt is much higher than the other components in the neutral fraction. The mixture was subjected to the chromatography and the DCM / methanol = 20/1 (v/v) was used as the eluent. Because the polarity of DCM / methanol = 20/1 (v/v) is much higher than the DCM/THF = 7/3 (v/v), the other neutral compounds were removed very cleanly. The GirT derivatives still retain on the silica gel and cannot be eluted due to its high polarity. Then a much higher polarity eluent, DCM / methanol = 3/2 was used to elute the GirT derivatives. The huge difference in polarity between these two parts ensure only AKs are present in the AKs fraction."

What about the cotinine behavior which is a well-known cigarette smoke tracer?

Response: Cotinine with a structure:

a neutral compound (lactam), but not an aldehyde or ketone compound.

It was eluted by DCM / methanol = 20/1 in the second step and not be study by this work.

Why the authors used paper filters? More adequate filters are quartz filter for which it can be considered that it cannot interact with the smoke sample.

Response: Thanks for the comment. That is true, we should use

the quartz filter. We bought the inappropriate filter, but we think this type filter will not affect the analysis results of this study. This is because the aldehyde and ketone cannot react with the filter; and the appropriate solvent was used to completely elute the compound from the filter, and the physical adsorption of the compound by the filter was excluded. The model of the Cambridge filter pad we bought is F319-04 (paper filters), which meets the requirements of Standard ISO3308:2000 (Routine analytical cigarette-smoking machine-definitions and standard conditions). But now the ISO3308:2000 has been withdrawn and revised by ISO 3308:2012. We will remove the model of Cambridge filter pad (F319-04) from this article so as not to mislead other researchers. In future research we will use the quartz filter. Thanks again.

Why collect the sample for 20 cigarettes in the same time. Interaction and chemical reaction may happen between the still collected particles and the compounds (in gas and PM phase). This may have a significant importance after the 20 smoking procedure.

Response: We are sorry that we have caused a misunderstanding.

This is the original sentence in the article: "The particle phase of MSS of 20 cigarettes was collected on whatman fiber pads." We mean that we totally collected 20 cigarettes' particle phase to do the next

experiment. In this procedure, five cigarettes are smoked consecutively under carefully controlled conditions. Then the particle phase on four fiber pads were combined. We have revised the text:

“20 cigarettes were divided into four groups on average. The particulate phase of MSS and SSS of each group was collected on fiber pads under a set of internationally agreed standard smoking conditions. The particle phase on four fiber pads were eluted by DCM/MeOH = 3/1, combined and stored at -20 °C.”

Details of the extraction step have to be given.

Response: According to your suggestion, we have added the following sentences:

“The original AKs are neutral compound, so first step the high-polarity acidic and basic fractions were removed by HCl and KOH modified silica gel (solid phase extraction method). In this step, the dichloromethane (DCM)/tetrahydrofuran (THF) = 7/3 (v/v) mixture was used as the eluent to elute the neutral fraction. The polarity of DCM/THF = 7/3 (v/v) is high enough to elute the aldehydes and ketones.”

“In the second step, the neutral fraction mixed with Girard reagent and the AKs are converted into corresponding GirT derivatives. The resulting GirT derivatives are very easy to separate. This is because the polarity of the quaternary ammonium salt is much higher than the other components in the neutral fraction. The mixture was subjected to the chromatography and the DCM / methanol = 20/1 (v/v) was used as the eluent. Because the polarity of DCM / methanol = 20/1 (v/v) is much higher than the DCM/THF = 7/3 (v/v), the other neutral compounds were removed very cleanly. The GirT derivatives still retain on the silica gel and cannot be eluted due to its high polarity. Then a much higher polarity eluent, DCM / methanol = 3/2 was used to elute the GirT derivatives. The huge difference in polarity between these two parts ensure only AKs are present in the AKs fraction.”

“1 g of anhydrous sodium sulfate, 2 g of KOH-modified silica gel, 1 g of anhydrous sodium sulfate, 4 g of HCl acid-modified silica gel and 1 g anhydrous sodium sulfate were sequentially packed into a 1 cm diameter column A. 10 g of ordinary silica gel, 1 g of anhydrous sodium sulfate and 0.5 g of KOH-modified silica gel were sequentially packed into another 1 cm diameter column B. The obtained cigarette smoke

particle phase (shown in Table 1) was dissolved in a 4 mL of DCM/THF = 7/3 (v/v) mixture, and the solution was subjected to the chromatography A using 25 mL of a DCM/THF = 7/3 (v/v) mixture as the eluent to elute the neutral fraction. The polarity of the mixed solvent is high enough to elute the aldehydes and ketones in the neutral fraction.”

The non-controlled evaporation of the solvent may lead to the evaporation of the more volatile cigarette smoke components.

Response: Agree. The solvent for neutral fraction and aldehyde/ketone fraction is THF/DCM and DCM, respectively. When evaporate the solvent by rotary evaporator, the bath temperature is always controlled below 10 °C to prevent some smoke components from volatilizing. Despite this, there is still some loss of volatile components. This is the drawback of chemical derivatization method compared to non-targeted method.

We have added the sentence in the revised manuscript:

“After removing the solvent below 10°C by a rotary evaporator, the neutral fraction was obtained.”

“Then the DCM layers were combined in a flask, the solvent was removed by a rotary evaporator below 10 °C.”

“The post-treatment process may result in the loss of volatile components, so some of small carbonyl specie cannot be observed in the Figure 4.”

The mass resolution of FT ICR MS measurement has to be given.

Response: According to your suggestion, we have added the sentence in the revised manuscript:

“The peaks with a 380000 resolving power at $m/z = 326$.”

Details of the “tentative identification and quantification” page 3 line 57/60. The quantification aspect is not discussed in the results and discussion section.

Response: According to your suggestion, we have added the following sentence in the revised manuscript:

“The quantitative results are listed in Table 2. Figure 4 and Table 2 shows that the AKPs in MSS and SSS are very close, but the content is different. In addition, the content of specific carbonyl is very different to each other. The content of high-carbon linear aldehydes (Henicosanal to Pentacosanal) in MSS is higher than that of SSS, but

the content of some low-carbon aromatic-contained aldehydes (trans-Cinnamaldehyde, alpha-Methylcinnamaldehyde) in sidestream smoke is higher than that of mainstream smoke. This may be due to the fact that MSS and SSS are formed in different ways of combustion. The higher temperatures and higher amounts of oxygen lead to the breaking of the carbon chain to form more linear aldehydes during MSS produced. The low level of oxygen in the production of SSS facilitates the pyrolysis process and form more unsaturated and condensed AKPs.”

The figures are too small and very hard to be clearly examined. What's mean MK and CK in Fig 6.

Response: According to your suggestion, we have change the size of the figures and revised the Fig 6.

Finally, authors have to give a comparison between what their findings are and what it is reported in the literature.

Response: According to your suggestion, we have revised the conclusion:

“In this work, the AKPs of MSS and SSS were successfully separated and analyzed in detail. Total 63 AKs were identified and

quantified by GCMS. Due to the complex post-treatment of chemical derivatization, a portion of volatile or unstable AKs are inevitably lost, so the number of AKs that characterized by GCMS is less than already reported.[52] Fifteen species of AKs were characterized by ESI FT-ICR MS and Orbitrap MS: O_{1-6} , N_1O_{1-4} , N_2O_{1-3} and N_3O_{2-3} . Regardless of the isomer, the total number of aldehydes and ketones obtained by ESI FTICR MS in MSS and SSS is about 1100 and 970, respectively. This number is much higher than the reported,[52] which indicated a large amount of component cannot be resolved by the GCMS. Furthermore, The nitrogen-containing carbonyls are more condensed and unsaturated than O_{1-3} carbonyls. The AKPs in MSS are more oxygenated and less unsaturated than that in SSS. Future work should focus on the toxicity of the isolated carbonyls to humans and the environment.”